# In vitro Modeling of Prion Strain Tropism

**DOI:** 10.3390/v11030236

**Published:** 2019-03-09

**Authors:** Etienne Levavasseur, Nicolas Privat, Stéphane Haïk

**Affiliations:** 1Institut du Cerveau et de la Moelle épinière, ICM, Inserm U 1127, CNRS UMR 7225, Sorbonne Université, 75013 Paris, France; etienne.levavasseur@icm-institute.org (E.L.); nicolas.privat@inserm.fr (N.P.); 2Assistance Publique-Hôpitaux de Paris, Cellule nationale de référence des MCJ, G.H. Pitié-Salpêtrière, 75013 Paris, France, AP-HP, Laboratoire de Neuropathologie, G.H. Pitié-Salpêtrière, 75013 Paris, France

**Keywords:** models, prion, tropism

## Abstract

Prions are atypical infectious agents lacking genetic material. Yet, various strains have been isolated from animals and humans using experimental models. They are distinguished by the resulting pattern of disease, including the localization of PrPsc deposits and the spongiform changes they induce in the brain of affected individuals. In this paper, we discuss the emerging use of cellular and acellular models to decipher the mechanisms involved in the strain-specific targeting of distinct brain regions. Recent studies suggest that neuronal cultures, protein misfolding cyclic amplification, and combination of both approaches may be useful to explore this under-investigated but central domain of the prion field.

## 1. Introduction

Prion diseases are a group of rare progressive neurodegenerative disorders that affect both humans and animals. Scrapie in sheep and goats, bovine spongiform encephalopathy (BSE) in cattle, and chronic wasting disease (CWD) in wild ruminants such as cervids are the most common forms of the disease in animals. Humans are mainly affected by sporadic Creutzfeldt-Jakob disease (sCJD), while genetic and acquired forms of the disease such as Gerstmann-Sträussler-Scheinker syndrome and variant CJD, respectively, are much less frequent. The agents responsible for prion diseases are essentially composed of the abnormally folded form (PrPsc) of the host prion protein (PrPc) [1]. Among patients with sCJD, a wide range of clinical and neuropathological phenotypes has been observed. The molecular basis of such phenotypic diversity involves a methionine/valine (M/V) polymorphism at codon 129 of the prion protein gene (*PRNP*), combined with different folding patterns of the PrPsc that accumulates in the brain of affected individuals. The latter is reflected by different sizes of the residual, protease-resistant core fragments (PrPres) of PrPsc, which suggests different PK cleavage sites in the conformations of the protein [2]. The most widely used classification distinguishes two main subtypes of PrPres (PrPres type 1 with a migration of the un-glycosylated form of the protein at 21 kDa and PrPres type 2 with a migration of the unglycosylated form of the protein at 19 kDa), as detected by the Western blot (WB) following digestion with proteinase-K (PK) [3,4,5]. Other researchers have proposed that an additional main type with an intermediate size of the un-glycosylated form can be detected in sporadic and iatrogenic CJD [6,7]. By correlating codon 129 genotype and PrPres types with clinical and pathological features in sporadic CJD, various molecular combinations corresponding to the most common phenotypic variants of sCJD were identified [4,8].

Although they contain no nucleic acids, different prion strains can be propagated in experimental models [9]. The distribution of the accumulation of PrPsc in brain regions varies with the strain, which determines the tropism phenomenon. It is associated with a lesion profile that affects the clinical form. Two strains were originally observed after inoculating goats with different scrapie isolates [10]. Many other strains are now acknowledged [11]. In cattle, three BSE strains have been identified so far [12,13,14], and it seems that at least two different strains are responsible for CWD [15,16]. In humans, a growing number of strains are identified using experimental models susceptible to human prions such as knock-in mice expressing human PrP, bank vole, and non-human primates [12,17,18,19]. Several in vivo studies have been conducted to address the phenomenon of prion tropism that is central to the definition of strain. A few decades ago, it was shown that brain lesions occurred in a region-specific manner in animals inoculated intra-cranially or by the peripheral route with three scrapie strains (87A, 31A, and 125A) [20]. The stereotactic inoculation of 139A, ME7, and 22L scrapie strains directly into five specific brain regions in C57BL/6, revealed that the efficiency of the strain replication may vary with the inoculation site [21]. In humans, the two main sCJD subtypes MM1/MV1 and VV2a have been recently associated with different prion strains, by inoculation to non-human primates and to knock-in mice [17,19,22]. They are characterized by different clinical phenotypes and neuropathological profiles (early dementia, myoclonus, isocortical involvement in MM1-MV1, and ataxia, late dementia, and cerebellar involvement in VV2a). On the other hand, vCJD, which results from the transmission of the agent responsible for classical bovine spongiform encephalopathy to humans [12,23], is associated with a particular involvement of the posterior thalamus [24]. Transmission experiments demonstrated that the same prion strain is associated with the vCJD cases observed in different countries [18].

Despite such evidence of a strain-specific tropism leading to preferential replication in given brain regions, the cellular mechanisms involved in this phenomenon have been marginally studied in the field. In an experimental hamster model of mink encephalopathy, different neuronal populations are targeted according to the strain (“hyper” or “drowsy”) that is inoculated into the sciatic nerve [25]. These first observations highlight the need for refined methods to address more precisely and specifically the strain tropism phenomenon. The recent development of cellular and acellular in vitro models of prion propagation has offered such an opportunity. In this case, we present and discuss the results obtained with primary neuronal cultures and protein misfolding cyclic amplification (PMCA). They provide the first evidence of a strain-specific neuronal tropism that may involve local molecular cofactors implicated in the PrP conversion process.

## 2. Cellular Models of Prion Strain Tropism

There are several prerequisites to investigate in vitro the strain-specific targeting of brain regions. First, from the point of view of cellular biology, a model permissive to different prion strains is mandatory. Research has long been limited by both the lack of available models and the poor infection level of cell cultures. Nonetheless, a number of assays have been designed by different laboratories (reviewed in Reference [26]). Highly susceptible sublines of N2a cells have been isolated and have proven useful as a rapid and sensitive alternative assay to the mouse bioassay for the detection of prions [27]. However, the scrapie cell assay remains restricted in its applicability since N2a sublines are resistant to prion strains of more immediate interest such as bovine or vCJD prions. In addition, immortalized cell lines derived from a malignant tumor are poorly relevant to study neuron-specific prion propagation that may vary with the precise neuroanatomical origin of the neuronal population. Based on primary cultures of cerebellar granular neurons from transgenic mice overexpressing ovine PrPc (tg338), a model of ovine prion propagation has been developed by Cronier et al [28]. Their results suggest that primarily grown cerebellar astrocytes (CAS) and cerebellar granular neurons (CGN) are permissive to PrPsc propagation. A similar system was subsequently developed with primary cultures of neurons from transgenic mice overexpressing human PrPc that were infected with a strain of sCJD, which was previously adapted to the same transgenic mouse line [29]. The results suggest that the anti-prion activity of three generic compounds (MS-8209, Congo red, chlorpromazine) observed in neuronal cultures is species-dependent or strain-dependent and recapitulates to some extent the activity reported in vivo in rodent models. Another in vitro model relies on cultured organotypic cerebellar slices (COCS) that can be infected with different prion strains (RML, 22L, and 139A) [30]. Prion-infected COCS reproduce the prion replication, inflammatory response, spongiform changes, and neurodegeneration observed in prion diseases.

Altogether, these studies suggest the feasibility of studying the cerebral tropism of prion strain using in vitro cellular models. To further decipher the mechanisms underlying the cerebral prion strain tropism, we set up primary neuronal cultures from cortex, striatum, and cerebellum of C57BL/6 mice. We showed that the kinetics of replication of three experimental scrapie strains stabilized in C57BL/6 mice (22L, ME7, 139A) differ with (i) the strain for a given model of cell culture, and (ii) the model of the cell culture for a given prion strain. Our results support the existence of a strain-specific neuronal tropism [31]. Furthermore, they show that a complex event such as the uncoupling of prion replication and toxicity that has been observed in mice [32] is reproduced in this model. The neurotoxic phase was initiated when a steady state of PrPres level was reached. This kinetics varied in a strain-dependent and neuronal-dependent manner. For instance, in granule cell cultures, the most cerebellar strain (22L) leading to granular cell loss in infected mice reached the plateau first with a more intense neuronal loss as compared to other strains.

In a second study, we propagated iatrogenic CJD (MM1 type), vCJD and sCJD (MM1) isolates in CGN cultures from mice overexpressing human M129 PrP, and a sCJD (VV2) isolate in CGN cultures overexpressing human V129 PrP [33]. Human prion propagation occurred at a later stage compared with CGN cultures infected with experimental scrapie strains [31]. In addition, different kinetics of prion propagation were observed between isolates in M129 PrP-CGN cultures suggesting a strain-specific neuronal tropism. Our results provided the first evidence supporting that human prion isolates could be propagated in primary cell cultures. This is an important step toward the search for chemical compounds targeting human prions, and the study of the cellular mechanisms involved in their brain distribution.

These different studies addressing the strain tropism phenomenon in cellular models share some limitations. (i) Primary cultures are usually established at embryonic stages, which might be an issue regarding the expression of cofactors. (ii) They are not fully representative of the cell populations present in the investigated brain region. (iii) The time-frame is limited and spontaneous degeneration and glial alterations may occur with time. (iv) A number of established models do not express an endogenous level of PrPc and have been transfected with the *PRNP* gene, which usually induces an overexpression of the protein.

In addition, developing a human cellular model susceptible to human isolates is a challenge for the years to come. Using a staggered exposure protocol, a recent study suggested that cultures of astrocytes derived from human induced pluripotent stem cells (iPSCs) are able to replicate CJD isolates in cells of human origin [34]. However, cultures of astrocytes may not be the most relevant model to address the strain tropism phenomenon. Although a heterogeneity of astrocytes based on morphological type is acknowledged [35]. Astrocyte cultures derived from iPSCs lack brain regional specificity. Developing human derived cell co-cultures including neuronal cells of different subtypes, astrocytes, and other cell types such as microglia [36] may provide useful models to study the complex interplay governing the differential strain-specific replication observed in some areas of the central nervous system.

## 3. Acellular Models of Prion Strain Tropism

Several laboratories have attempted to develop acellular systems reproducing in vitro the conversion of PrPc into PrPsc observed in vivo [37]. In 2001, the development of the protein mis-folding cyclic amplification (PMCA) opened the way for in vitro production of infectious prion proteins [38,39,40,41]. This system allows an exponential amplification of PrPsc using cyclic sonication and incubation, which is conceptually analogous to the amplification of DNA by PCR (Scheme 1).

In addition, the PMCA products retain the main characteristics of the original PrPsc such as electrophoretic mobility, glycosylation pattern, and resistance to proteinase K [38,39,40,42,43]. Importantly, PrPsc generated by PMCA is infectious when inoculated to wild type animals [39,41,42]. PMCA allows studies on genotypic and species barrier of prion transmission [44,45,46,47,48,49,50,51,52], the detection of PrPsc in various biological fluids (blood, urine, saliva, LCR) [53,54,55,56,57,58], de novo generation of prions [59,60,61], and the identification of cofactors involved in PrP conversion [59,61,62,63,64,65,66,67,68,69,70,71,72,73]. To overcome some of the above-mentioned issues specific to cellular models, and to assess whether strain tropism can be studied in an acellular system, we recently developed a region-specific PMCA (rsPMCA) [46]. This technique allowed the use of various animal and human tissues, prepared from adult individuals. In addition, the tissue lysates used as substrates contain all the molecular factors of the different cell types and the extracellular space that are present in each brain structure (Scheme 2). The development of rsPMCA intended to assess whether prion strain tropism involves region-specific molecular factors.

We first compared the distribution of PrPres in five brain regions from C57BL/6 and tg650 (M129) mice intra-cerebrally inoculated with three different scrapie strains and vCJD, respectively, with the level of amplification by rsPMCA using the same regions taken from healthy animals as substrates [46]. Our results showed that rsPMCA partly matched, in the murine context, the regional targeting observed in vivo with 139A, ME7, 22L, and vCJD strains. Then, using normal human brain tissues from subjects with an MM or VV genotype at codon 129 as substrates, we confirmed these results with PrPsc from vCJD and sCJD VV2 isolates as seeds. When the results obtained using animal and human substrates were pooled, a highly significant correlation between in vivo tropism and in vitro conversion efficiency was obtained, which suggested that the mechanisms involved are common to various prion strains. A significant correlation was maintained when the relative amplification values were corrected for the PrPc relative level of each substrate, which suggests the involvement of region-specific cofactors different from PrPc. This was confirmed by showing that tissue preparations from mice devoid of PrPc could modulate PMCA efficacy in a tissue-specific manner [46].

In contrast with our results, the PrPsc in vitro conversion efficiency did not match the pattern of deposition observed in a hamster scrapie model [74]. The discrepancy between the two studies may be simply explained by the use of distinct prion strains in different species. It is worth noting that the effect of PMCA cofactors may vary with species. Deleault et al. [69] reported that, whereas hamster PrPsc preferentially utilizes RNAs as a cofactor, RNAs fail to facilitate mouse PrPsc amplification. In addition, a clarification step of the substrates using centrifugation was performed in the hamster study by Hu and collaborators [74] and may have removed some of the components responsible for region-specific modulation of PMCA. It was suggested that in vitro conversion using a hamster substrate was mostly dependent on the availability of PrPc. However, it was not confirmed in a study using grey and white matter of the same sample [75]. Moreover, it was shown that a high level of PrPc expression does not systematically correlate with a high conversion rate. Cell-free in vitro conversion activity assay using brains from PrP overexpressing mice as substrate does not show a proportional increase in conversion activity as compared with wild-type mice [76]. Our data obtained using PMCA suggest that additional molecular factors, distinct from PrPc, are involved in the regional brain targeting by human prion strains. Our conclusion is supported by in vitro results showing that PrP expression is not the limiting factor between highly and poorly susceptible N2a cells [77].

The glycosylation of PrPc is known to interact with prion propagation [78]. Moreover, the glycoform ratio of PrPres may vary between brain structures in CJD [79]. A study has demonstrated that brain targeting following peripheral inoculation of two TSE strains in transgenic mice expressing different glycosylated forms of PrP was profoundly influenced by the glycosylation status of host PrP [80]. Intracerebral inoculation of three TSE strains in mice, carrying mutations at the first, second, or both PrP N-linked glycosylation site, suggests that alterations in PrP glycosylation modulate the incubation period, and that glycosylation at the first site may have an effect on strain targeting [81]. In our work, the glycosylation status of PrPc did not appear to be an important region-specific cofactor since we did not observe any significant difference in the distribution of PrPc glycoforms, according to the human brain regions used as PMCA substrates and showing various efficiencies in PrP conversion [46].

Among the non-PrP molecular factors relevant in PMCA, ribonucleic acids are potential candidates [59,62,65]. A study with murine strains showed that RNAs are catalysts of the strain-specific conversion [82]. The use of murine and hamster substrates depleted in RNA altered the rate of strain adaptation [83]. In our hands, the digestion of vCJD seeds and tgMet brain substrates with RNAse A, T1, V1, and S7 nuclease or benzonase prior to rsPMCA did not alter the amplification differences between the brain regions used as substrates. This suggests that RNAs are not essential cofactors that influence brain targeting by human prion strains. Other cofactor candidates include metal ions [64], glycosaminoglycans [84], laminin receptor (LRP/LR) [85,86,87], and anionic lipids [61,75]. For example, poly-anionic cofactors affected the strain specificity of infectious recombinant prions generated in vitro [88]. Another group has reported the faithful and stable replication in vitro of a hamster strain using also recombinant PrP as a substrate, and a mixture of phospho-lipoproteins and synthetic nucleic acid polyA [89]. It is worth noting that a possible cause of divergence between the classifications of sCJD proposed by Collinge and Parchi may be related to the presence of metal ions such as copper and zinc [90], which would affect PrPsc conformation.

## 4. Combining Cellular with Acellular Models

Thus, the use of PMCA allows to selectively investigate the role of different substrates. Most of them originate from experimental models that overexpress PrPc. When available, human substrates are valuable. However, the amplification efficiency is affected in our hands by the post mortem delay. Some research teams have attempted to develop more simple substrates. The use of fractions containing lipid rafts where PrPc content is enriched and isolated from hamster brain homogenates as substrates has been helpful to study the components (including cofactors) involved in PMCA reaction [68,91]. Others have used purified PrPc from the brain [59], from cultured cells [92], platelet lysates [66], or recombinant PrP expressed in bacterial cells as substrates [61,93], and have proposed minimal components that are necessary to produce de novo infectious prions with PMCA. Using, for the first time, lysates of cultured mammalian cells as substrates, Mays and collaborators showed that in vitro amplification of mouse-adapted and hamster-adapted strains is possible [94]. This was an important step because cell culture lysates were considered until they were unable to support PrP conversion in PMCA alone [95]. However, they had been previously used in a cell free assay to investigate the role of polyanions in the process of PrPsc formation [76]. Furthermore, Yokoyama and collaborators showed that vCJD prions could be amplified using human cell lysates (293F cell line) supplemented with heparin as substrate, and that the effect of heparin on cell-PMCA was strain dependent [96].

The perspective of using cell lysates with PMCA is promising because these substrates are much less expensive than brain material from animal models, which is easily available and more ethical. We confirmed that cell lysates could be used as PMCA substrates using N2a cells (data not shown). The next step is to set up a neuron-specific PMCA based on primary cultures of neurons from various brain structures such as the mouse cortex, the striatum, and the cerebellum as substrates to study the strain-targeting phenomenon (Scheme 3).

Although a statistical difference was detected only between 22L and ME7 strains, using CGN lysates from C57BL/6 mice as substrates for nsPMCA, our preliminary data suggest that amplification is faster with 22L, intermediate with 139A, and slower with ME7 strains (Figure 1). The kinetics of infection in CGN follows the same order, i.e., 22L is the first strain to reach its steady state of PrPres accumulation, while 139A was the second and ME7 was the third strain to reach a plateau [31]. Moreover, when considering previous in vivo results, the incubation period in C57BL/6 mice inoculated in the cerebellum region with these three strains was the shortest with 22L, intermediate with 139A, and the longest with ME7 [21].

Our preliminary results suggest that primary neurons can be used as a substrate to explore prion tropism in acellular models. Our primary neuronal cultures are devoid of microglial cells. It has been shown, however, that microglia contribute to amyloid formation and enhance neuronal destruction in prion diseases [97]. On the other hand, the extensive microglial activation accompanying prion diseases represents an efficient defensive reaction [98]. Our combined approach also allows the study of the role of glial cell-associated molecular factors in PrPc conversion by modifying the ratio of microglial cells and astrocytes in co-cultures, and by adding conditioned medium from prion exposed microglia to neuronal cultures. In addition, manipulating cell components in neuronal cultures from various brain areas (using selectively lipases, proteases, etc.) will help us identify potential cofactors. As a complementary approach, the use of antibodies or competitors and the manipulation of gene expression should allow us to target most known PrPc partners [99]. These strategies may be completed by PrP interactome studies that should take into account possible variations between different neuronal cultures. In the hypothesis that cofactors would be revealed in these cellular or acellular systems, KO models for these newly identified molecules will be helpful to validate their role in modulating the prion-strain tropism.

## 5. Conclusions

Deciphering the mechanisms involved in the strain tropism phenomenon requires new approaches relying on in vitro models of prion replication. Recent evidence from experiments using primary infected cultures supports that a strain-specific tropism for different types of neuronal populations may contribute to the differential targeting of brain areas. This phenomenon may be complementary to the role of intercellular connectivity in the spread of misfolded prion protein aggregates. Region-specific PMCA suggests that such a neuronal tropism involves molecular factors in addition to the potential role of cell functions and cell-to-cell interactions that may differ between cells from various brain areas. These results underline the value of a regional in vitro study of prion propagation that could be refined by the use of co-cultures of neuronal and non-neuronal cells. The combination of these approaches should make possible the identification of inhibiting or enhancing local cofactors of prion conversion that are involved in strain tropism. Beyond the prion field, these new methods may also help us understand the selectivity of brain lesions in non-prion neurodegenerative diseases, in which a protein-specific pattern of lesion dissemination and the occurrence of prion-like mechanisms of propagation have been shown [84].

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
