# Peer review of "In vitro Modeling of Prion Strain Tropism"

_viruses, 2019, doi:10.3390/v11030236_

Round 1

Reviewer 1 Report

This is a timely and valuable review of in vitro modelling of prion strain tropism. I have only a few minor suggestions for improvement:

1. L72: “grain” should be “granular”, I assume.

2. L137: “authorized” should be “allowed”

3. L215: It would be valuable and appropriate here to mention the earlier use of human platelet lysates as a PMCA substrate reported by Mark Head's group in a couple of papers.

Author Response

We thank the Reviewer for his comments.

We have modified accordingly the text as proposed (new numbering with hidden revision marks) line 87 and 162 and quoted the reference for the use of human platelet lysates as suggested (Jones et al, 2009), line 245.

Reviewer 2 Report

The manuscript entitled ‘in vitro modelling of prion strain tropism’ by Levavasseur et al highlights the emerging significance of cell and cell-free models to our understanding of prion strain.

A strength of the review is the authors expertise in this area of prion biology. This is highlighted by a significant proportion of the manuscript reporting the results of their own work. This is somewhat balanced by reference to other work in the area of in vitro modelling.  However, the review would be of more value to readers if a more detailed account of the work of others was also described in the review. For example, a more detailed explanation of the influence of cofactors present in the substrate used for PMCA (references 50, 52-66 and more recent publications from the groups of Baskakov and Castilla) would provide an excellent introduction to the significance of the authors own work performing region specific PMCA.

The authors introduce the potential significance of human prion disease using the example of sporadic CJD and in particular describe the PrPres types identified by Parchi et al.  A balanced review would also acknowledge the PrPres types identified by Collinge et al. This is particularly relevant in light of the focus of the review in part on cofactors.  A possible explanation for the different PrPres types identified by Parchi and Collinge is the presence of metals as cofactors, described by Wadsworth et al 1999, which could be affected by the composition of cell-free substrates.

The authors claim the first use of cell lysates as substrates in the PMCA by Mays 2011, but it should be acknowledged that cell lysates had been used in cell-free assays prior to this as reported by Lawson et al 2010.

Make it clear that Scheme 2 is a model for rsPMCA and does not represents the results of their work.

It would be helpful if Figure 1 could include an image of the results described in reference 21 (with the publisher’s permission).  The correlation is overly simplified, as the cell-based data is based on kinetics (how fast infection occurs), whereas the PMCA data presented is how much.  It would be useful to also include the incubation period in mice as reported in reference 21, which suggests 139A faster than ME7 faster than 22L, which does not correlate with the reported kinetics in cell lines or amount of PrPres produced in the PMCA.

Please provide statistical analysis of the data presented in Figure 1.

Minor 

·     Line 72 is ‘cerebellar grain neurons’ correct

·     Line 93, sentence commencing ‘For instance,’ is poorly worded. Granule cells would be more appropriate to granular cells.

·     Line 113 correct ‘protocole’ to protocol

·     Line 206 it is not clear what is meant by the term ‘Criss-cross’ 

·     Line 263 change ‘allows to study the role of’ to ‘allows the study of the role of’ or ‘allows the study of’

·     Line 265 change ‘exposedmicroglial-cultures’ to ‘exposed microglial-cultures’

·     Line 270 change ‘interactomes’ to ‘interactome’

·     Line 285 change ‘involved in the strain’ to ‘involved in strain’

·      Line 129 Provide a reference/s for the statement ‘In addition, the PMCA products

·      retain the characteristics of the original PrPs such as electrophoretic mobility, glycosylation pattern (amino acid composition), resistance to proteinase K, heat resistance, resistance to denaturation by guanidine hydrochloride’.

·     Line 137, it is not clear what is meant by ‘authorized’

·     Line 176, include the reference number for Hu.

·      Line 179, reference 68 concludes ‘Cell-free prion protein misfolding by sporadic CJD subtypes was affected by brain tissue composition and not related to absolute levels of PrPC’ not as stated by the authors ‘mostly dependent on the availability of PrPC’

·     Paragraph beginning line 261, it is not clear what the relevance of this paragraph is in the context of the preceding paragraph, where the focus was on differences in neuronal populations. Consider revising introduction of this paragraph to better introduce this important topic.

Author Response

We are grateful to the Reviewer for his careful reading and the numerous comments and suggestions that help to improve our manuscript in terms of accuracy and clarity.

We have consequently made the following modifications (please consider the line number of the revised manuscript when revision marks are hidden):

·         Regarding the influence of cofactors described by others, several references have been added and detailed line 224 (Katorcha et al, 2018), and lines 229 to 233 (Fernandez Borges et al, 2018; Makarava et al, 2018).

·         The different existing sCJD classifications have been introduced, line 35. The role of metal ions as cofactors and as possible source of divergence between Parchi’s and Collinge classification has been mentioned, starting line 233.

·         The reference to Lawson’s study has been properly introduced and explicited, starting line 250.

·         Regarding the Scheme 2 (now scheme 3) line 262, we would like to emphasize that it represents a general procedure for the use of PMCA based on specific neuronal cultures (nsPMCA), and not on rsPMCA (region-specific PMCA) as indicated by the Reviewer. Although we have started to work following this general proposed methodology, the data indicated on Scheme 3 (cell culture and PMCA) are purely indicative and do not necessarily represent our own results. To clarify this point, we have accordingly added this mention at the end of the Scheme 3 legend.

·         Considering the relatively short imparted time given by the editor, we could not ask the publisher’s permission to include an image of the ref “Hannaoui et al, 2013” as requested.

·         We agree with the Reviewer when pointing out the overly simplified correlation between cell-based data and PMCA data. We therefore reformulated our observations (starting line 272).

·         Similarly, line 277, we added and discussed the incubation periods of 139A, 22L and ME7 strains, originally described by Kim et al, 1987, in relation with PMCA and cell culture results. However, we are surprised by the Reviewer conclusion, since we find homogeneity in all three models (PMCA, cell culture, and inoculated mice). It is possible that the wrong brain region was considered in Kim’s paper, for example caudate nucleus or thalamus, instead of cerebellum, which is the region of interest in our study.

·         As requested, we included a short indication on the performed analytical test at the end of the Figure 1 legend, line 291. We want to stress out the point that these results refer to preliminary data that remain to be repeated and confirmed. Thus, we moderated our assertion regarding the usefulness of PMCA with cell lysates as substrates, line 272.

·         Line 87 “cerebellar grain neurons” was corrected to “cerebellar granular neurons”

·         Line 108, “granular cells” was replaced by “granule cells”.

·         Line 128, “protocole” was corrected to “protocol”

·         Line 237, we replaced the term “Criss cross” by “Combining”

·         Line 300, we corrected to “allows the study of the role of…”

·         Line 302, we corrected the sentence as follows: “adding conditioned medium from prion exposed microglia to neuronal cultures”

·         Line 307, “interactomes” was corrected to “interactome” as requested

·         Line 322, it was corrected to “involved in strain”

·         Line 154, we have included several references supporting the characteristics (that we have somewhat shortened for the sake of readability) of amplified PrPsc.

·         Line 162, we replaced “authorized” by “allowed”.

·         Line 200, the “Hu et al” reference was properly added.

·         Line 203, we agree with the mistake pointed out by the Reviewer, and we modified accordingly the text.

·         Line 296, we clarified the paragraph as requested, and established a connection with our previous results.

Reviewer 3 Report

The review discusses research programs addressing prion strain tropism, a fascinating subject with great practical implications. The important research goal is to identify tissue-specific cofactors responsible for differential propagation of distinct prion strains. 

I find the article well organized but writing a bit hard to follow.  For example:

Line 45: “The variant of CJD” should simply be “Variant CJD”.

Line 88: “cultured cells” would be better as “cell culture”.

Line 265: “using or not transwell systems” sounds strange.

Line265: “prion exposedmicroglial-conditioned” sounds very strange.

Some style improvement would greatly help the reader and boost the possible impact of the article.    

Author Response

We thank the Reviewer for his comments.

We have made the corrections as suggested (please consider the line number of the revised manuscript when revision marks are hidden).

Line 59, we corrected to “On the other hand, vCJD… »

Line 103, we corrected to “cell culture”.

Line 302, we have removed the « transwell system » mention, and rephrased the « prion exposed microglia » part.

The general style has been improved, thanks to the referees’ comments.

Reviewer 4 Report

In this review, the authors discuss the use of cellular and acellular models to decipher the mechanism involved in the strain-diversity of prions in distinct brain regions. The figures included in the submission are depicting the different usage of PMCA using brain and cellular samples.

For readers that are new to this topic, the authors could provide a short background on the prion strains diversity (eg. in scrapie or CJD or BSE), possibly with a figure showing the immunoblotting bands underlying different prion strains (eg.CJD).

Similarly, the authors can include a figure showing the principles of PMCA before discussing the application of this technique in classifying prion strains.

Author Response

We thank the Reviewer for his comments.

As suggested, we enriched the introduction with a paragraph about the diversity of prion diseases in human and animals.

We also included, line 146, a figure (Scheme 1) illustrating the mechanism of protein misfolding cyclic amplification, for the non specialist.